# Factors influencing survival outcomes in patients with stroke at three tertiary hospitals in Zimbabwe: A 12-month longitudinal study

Farayi Kaseke[1][☉]*, Lovemore Gwanzura[2‡], Cuthbert Musarurwa[3‡], Elizabeth Gori[4‡], Tawanda Nyengerai[5‡], Timothy Kaseke[6☉], Aimee Stewart[7]

1 Department of Physiotherapy, School of Health Sciences, College of Medicine and Health Sciences, University of Rwanda, Kigali, Rwanda, 2 Department of Laboratory Diagnostic and Investigative Sciences, Faculty of Medicine and Health Sciences, University of Zimbabwe, Harare, Zimbabwe, 3 Department of Biomedical Laboratory Sciences, School of Health Sciences, College of Medicine and Health Sciences, University of Rwanda, Kigali, Rwanda, 4 Department of Medical Biochemistry, Molecular Biology and Genetics, School of Medicine and Pharmacy, College of Medicine and Health Sciences, University of Rwanda, Kigali, Rwanda, 5 The Best Health Solutions, Johannesburg, Gauteng, South Africa, 6 Zimbabwe AIDS Prevention Project (ZAPP), Harare, Zimbabwe, 7 Department of Physiotherapy, University of The Witwatersrand, Johannesburg, South Africa

☉ These authors contributed equally to this work.
‡ LG, CM, EG and TN also contributed equally to this work.
* farayi.kaseke@gmail.com

**Data Availability Statement:** All relevant data are within the manuscript and its Supporting Information files.

## Abstract

### Background

In this longitudinal study, we aimed to determine factors influencing survival outcomes among patients with stroke at three tertiary hospitals over a 12-month period. The investigation sought to uncover influential determinants to enhance the precision of prognostic assessments and inform targeted interventions for individuals affected by stroke.

### Methods

Employing a longitudinal study design, participants were observed for 12 months from baseline, censoring survivors at the endpoint. The dataset originated from a comprehensive study involving stroke patients treated at three referral hospitals in Zimbabwe: Parirenyatwa, Sally Mugabe, and Chitungwiza Central Hospitals. The primary outcome variable, the duration of survival until death, was measured in days from the initiation of stroke treatment. Gompertz parametric regression analysis was utilized for data modeling following AFT model diagnostics.

### Results

In our study, 188 stroke patients were enrolled at baseline. However, 51 patients were excluded from the analysis due to either missing information or loss to follow-up. Among the remaining 137 patients who were tracked over a 12-month period, 42% were censored, and 58% were deceased. Individuals utilizing 'Free Service (older than 65/pensioners/retirees/ social welfare)' hospital bill payment methods showed a decreased risk of death, (adjusted

**Funding:** Farayi Kaseke was supported by the Training for Research Excellence and Mentorship in Tuberculosis (TRENT) Program, Grant Number D43TW009539. from the National Institute of Health (NIH) as a PhD scholar. This grant was awarded in collaboration with the University of Zimbabwe Faculty of Medicine. The funding was used for data collection purposes. The funding grant ended on 31st December 2023.

**Competing interests:** The authors have declared that no competing interests exist.

hazard ratio; aHR: 0.4, 95% CI: 0.20, 0.80), suggesting a protective effect compared to cash paying patients. Those who had attained a secondary school level education displayed a significantly lower risk of death (aHR: 0.4, 95% CI: 0.24, 0.79) compared to those with primary level education. Age was a significant risk factor, with individuals aged 45–65 and those over 65 years showing higher adjusted hazard ratios 3.4 (95% CI: 1.42, 8.36) and 3.7 (95%CI:1.44, 9.36), respectively, relative to those below 45 years of age. Housing status revealed a protective effect for those residing with parents/relatives (aHR: 0.4, 95% CI: 0.20, 0.64). Total functional outcome demonstrated significantly lower hazards for individuals with mild or moderate (aHR: 0.2, 95% CI: 0.09, 0.40) and severe outcomes (aHR: 0.2, 95% CI: 0.10, 0.46) compared to those with very severe outcomes.

## Conclusion

The study findings demonstrate that hospital bill payment methods, housing status and staying with relatives, educational attainment, functional outcome, and age significantly affect survival outcomes among stroke patients. This highlights the need to consider socio-demographic and clinical variables in the development of prognostic assessments and targeted interventions for individuals recovering from stroke.

## Introduction

Stroke, a significant global health challenge, is a major contributor to disability and mortality, particularly affecting low- and middle-income countries, where approximately 70% of stroke-related deaths and 87% of stroke-related disabilities occur [1, 2]. Within sub-Saharan Africa (SSA), the burden of stroke is disproportionately high, contributing significantly to stroke-related deaths and disability rates [2–4]. Alarming statistics reveal an annual incidence rate of up to 316 per 100,000, a prevalence of up to 1460 per 100,000, and a staggering 3-year mortality rate exceeding 80% [1, 4].

Challenges in managing stroke in the SSA region are further emphasized by Sahle Adeba et al. [5] who highlighted a high incidence of mortality among adult stroke patients. Understanding factors influencing survival outcomes, will provide a foundation for targeted interventions customized to the region's unique socio-economic and healthcare landscape. Age consistently emerges as a pivotal factor influencing stroke outcomes, with individuals over 65 years facing a higher risk of death [6, 7]. Furthermore, other studies highlighted the predictive role of old age in stroke incidence, emphasizing the importance of age-related considerations [8, 9]. In addition, lower educational attainment is associated with a heightened risk of mortality from brain stroke [10, 11]. Unemployment, another socio-economic factor, has also been linked to increased mortality risk among stroke patients [12].

Recognizing the nature of stroke outcomes, it is important to delve into additional determinants such as hospital bill payment modalities, educational attainment, housing status, and total functional outcomes. Hospital bill payment modalities can reveal economic barriers and financial stress that affect recovery trajectories post-treatment [12, 13]. Educational attainment may influence health literacy, access to healthcare resources, and ultimately, health outcomes [14, 15]. Housing status correlates with social support levels and environmental stability, which are pivotal for long-term recovery and well-being [16]. Total functional outcomes provide a measure of recovery, encompassing physical, cognitive, and social dimensions [17]. This

study sought to contribute to this understanding, by exploring factors influencing survival outcomes among stroke patients in Zimbabwe. By doing so, we aimed to inform targeted interventions and enhance the overall management of stroke in this high-burden region, aligning with global efforts to reduce the impact of stroke on communities worldwide.

## Materials and methods

### Study design

Our study employed a longitudinal design that initially enrolled 188 stroke patients at baseline. The data were collected prospectively. The researcher and research assistants identified eligible patients based on the inclusion criteria and consecutive sampling was done with patients recruited as they were admitted from each of the three hospitals.

Fifty-one patients were excluded from the analysis due to missing information or loss to follow-up during the course of the 12 months. The exclusion of these 51 observations was intended to improve the robustness and reliability of the analysis by focusing on participants with complete and reliable data. The subsequent analysis included the remaining 137 patients who were followed-up for 12 months, from baseline, after which the surviving participants were censored (Fig 1). The initial sample size calculation accounted for anticipated attrition rates, to maintain the study's statistical power despite the loss. This consideration was to ensure that the conclusions drawn were based on a more homogeneous and complete dataset. The manuscript is a sub-analysis of dataset derived from a larger study encompassing stroke patients initiating treatment at three referral hospitals (Parirenyatwa, Sally Mugabe, and Chitungwiza Central Hospital), in Harare, Zimbabwe [18]. At all the three hospitals, the study participants were admitted in the stroke units or medical wards. At the time of data collection, only Parirenyatwa Group of Hospitals had a stroke unit. Parirenyatwa and Harare Central Hospitals were manned by neurologists while Chitungwiza Central hospital was manned by

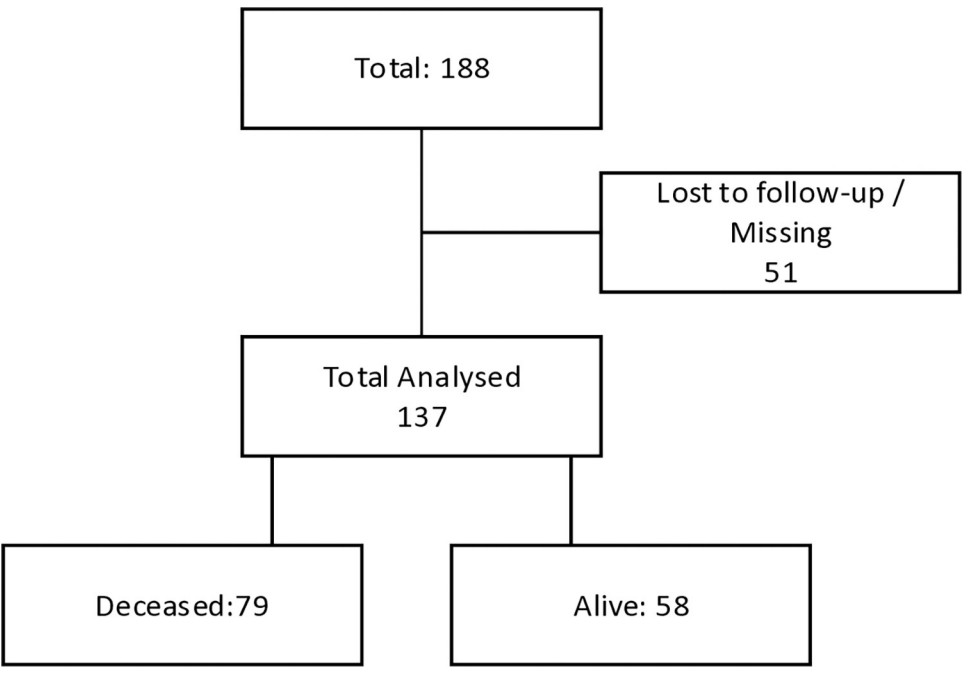

**Fig 1. Patient status overview.**

general practitioners and physicians who were not specifically neurologists. Weekly outpatient stroke clinics for "all" stroke patients discharged from hospitals are run at these hospitals. However, due to shortage of physiotherapists and occupational therapists manning the outpatient stroke clinics, patients may receive therapy only once or twice a month.

## Inclusion and exclusion criteria

All patients (age $\geq$ 18 years) diagnosed with stroke and admitted to the stroke units or the medical wards of the three public hospitals during the study period were included. The study excluded admitted patients who died before medical stabilization or refused to participate, as well as those who had a Transient Ischaemic Attack (TIA). The rationale for excluding TIA cases was to ensure the study population consisted of patients with more definitive and severe cerebrovascular events, thereby reducing variability and improving the precision of the study's outcomes.

## Study population

The source population was all stroke patients with a clinical diagnosis admitted at the three public referral hospitals in Harare, Zimbabwe from July 2015 to Nov 2017. Not all the patients had a confirmed stroke diagnosis based on a CT scan since some could not afford the cost. One hundred and eighty-eight adult stroke patients were included in the study.

## Study variables

The primary outcome variable was the duration of survival until death, measured in days from the initiation of stroke treatment to the occurrence of death or censorship. Various independent variables were examined for their potential influence on the survival time of stroke patients, including sex, hospital of admission, age group, marital status, occupational status, level of education, housing status, cigarette smoking status, hospital bill payment modality, transport modality, stroke side, comorbidities, time of stroke, total functional outcome, and dependency. Free service in this study referred to those on social welfare, older adults over 65 and pensioners/retirees. This set of variables allowed for a thorough exploration of factors contributing to the survival outcomes in the study population. Analysis by Human immunodeficiency (HIV) status was not performed due to a significant proportion of participants with missing HIV status data (71 cases—52%), but remained included in the analysis of other variables. The exclusion of HIV status was necessary to avoid misleading conclusions, as 69.01% of the cases with missing HIV data were deceased, which could have introduced bias if incomplete or imputed data had been used.

## Operational definitions

**Stroke:** Stroke was defined as a focal neurological deficit caused by a disturbance of blood circulation to the brain following an acute or sudden disturbance of brain function vascular in origin causing disability lasting more than, or leading to death within 24 hours. The study included both ischemic and hemorrhagic stroke patients.

**Event**: Death recorded among stroke patients during 12 months of follow-up (in the hospital or after hospital discharge in the defined period and reported by a caregiver).

**Censored**:(i) If the patient was alive until the 365-day post stroke,

**Time to event**: Time from hospital admission until the death of the patient was confirmed either during admission or after discharge within the 12 months of follow-up.

**Twelve months case fatality rate**: Was calculated using the number of deaths due to stroke during the 12-months follow-up period as the numerator and the total number of stroke patients during the 12 months follow-up period as the denominator.

## Data collection procedure

Data collection was carried out by one research assistant for each hospital who was trained in the use of the data collection tools. The research assistants collected all relevant data from patients' charts and interviewed the patients/caregivers using a validated data extraction sheet and questionnaire. The principal investigator was present to conduct data quality checks upon collection by the research assistants. The data collection tool was validated for reliability before deployment. The Functional Independence Measure (FIM™) tool also used in this study, was validated and found to be a reliable tool for stroke patients [19–21]. Participant history was recorded from the patient and/or relatives using a language they were comfortable communicating in (Shona or English). Clinical information data was abstracted from the patients' notes or files. Data on socio-demographics (age, sex, education level, occupation, marital status etc), clinical data (clinical presentation, past medical illness, duration of symptoms, CT-scan etc) were collected by interview at baseline, while death was recorded upon follow up at 3 months or at 12 months. Finally, the participants were assessed for their overall functional outcomes using the FIM tool at baseline and at follow up. Total score for FIM was measured as the sum of the motor and cognition subscale scores. This was transformed into ordinal categories (mild/moderate, severe, and very severe) to generate proportions for each category.

## Data analysis

The statistical analysis was conducted using STATA software version 15.1 (StataCorp, College Station, TX). Descriptive statistics were expressed as frequencies with corresponding percentages. The Kaplan Meier survival curves were utilized to compare patients by covariates. Bivariate Cox proportional hazards regression models were fitted for each explanatory variable. Variables with p≤0.25 in the bivariate Cox proportional hazards analysis were deemed eligible for further analysis. However, the Cox proportional hazards model failed the model diagnostics, as indicated by the global test result's significance (p<0.05) and a systematic departure from a horizontal line in the plot of scaled Schoenfeld residuals against transformed time, revealing violations of proportional hazard assumptions. Consequently, Gompertz parametric regression analysis was employed for modeling the data post accelerated failure time (AFT) model diagnostics. Variables with a p-value < 0.05 were considered significantly associated with the outcome in the multivariate analysis, and the adjusted hazard ratios (HR) with a 95% confidence interval (CI) were utilized to express the associations between the outcome and independent variables. The Wald test was used to retain variables that had a p-value < 0.05, utilizing a process of iteratively removing, refitting, and verifying until all significant variables were retained in the final main effects model.

## Accelerated failure time model selection and diagnostics

The AFT models were used since the Cox proportional hazards (PH) assumptions were violated. For an in-depth exploration of the impact of candidate covariates on the survival time of stroke patients, bivariate analysis was conducted for each covariate using different baseline distributions in AFT models. The multivariable analysis incorporated the Exponential, Gompertz, Weibull, and log-normal distributions for the baseline hazard function, considering the most significant covariates. To assess the goodness of fit, estimated cumulative hazard plots were plotted against Cox-Snell residuals, providing insight into how well the AFT models captured

the stroke dataset. Furthermore, the Akaike's information criterion (AIC) was employed to assess the model's goodness of fit and guide the selection of the most appropriate AFT model for the analysis.

## Ethics approval and participant consent

Permission to carry out this study was granted by the institutional review boards at the three referral hospitals Parirenyatwa, Sally Mugabe (formerly Harare Central) and Chitungwiza Central hospitals. Ethical approval was given by the Joint Parirenyatwa Group of Hospitals and University of Zimbabwe Research Ethics Committee (JREC– 312/12) and The Medical Research Council of Zimbabwe (MRCZ– 34/78). All participants gave written informed consent and in the event that the participant could not communicate, assent was given by the caregiver.

## Inclusivity in global research

Additional information regarding the ethical, cultural, and scientific considerations specific to inclusivity in global research is included in the S1 File.

## Results

There was substantial loss to follow-up. Fig 1 compares those who were lost to follow-up versus those who remained in the study for analysis.

## Sociodemographic characteristics

Table 1 presents the sociodemographic and clinical characteristics of stroke patients followed for 12 months. In total 79 (58%) participants were deceased and 58(42%) were censored. Sex distribution showed 83(61.3%) females and 54(38.7%) males, with corresponding deceased percentages of 57.1% and 58.5% for each sex. Hospital of admission showed statistically significant variations in mortality (p = 0.005), with Parirenyatwa having 33/47 (70.2%) of its patients with stroke deceased, Sally Mugabe Hospital with 22/54 (40.7%), and Chitungwiza Central Hospital with 24/36 (66.7%). Age group categorization revealed significantly higher mortality (p<0.001) in those over 65 years (80.8%), and marital status showed no statistically significant variation with 58.2% deceased among married individuals and 56.9% deceased individuals in the 'Not in Union' category, which included the single, widowed, divorced, and separated individuals (p = 0.876). The unemployed category had 61.3% deceased, while among retirees 73.1% of the participants died and there was a significant variation in the proportion of deceased participants based on employment status (p = 0.017). Educational attainment also displayed a statistically significant variation in mortality with 78.3% deceased among those with only primary school education and 36.9% deceased among those with secondary school education (p<0.001).

  **Sociodemographic characteristics including loss to follow-up data.** Table 2 shows the sociodemographic characteristics of stroke patients, including their survival status and loss to follow-up/missing data. Patients admitted to Sally Mugabe Hospital had the highest survival rate (53.3% censored) and the lowest loss to follow-up (10.0%), while those from Parirenyatwa and Chitungwiza Central Hospitals experienced higher mortality rates (41.3% and 50.0%, respectively), with a higher proportion of patients lost to follow-up at Parirenyatwa (41.3%). Older patients (>65 years) had the highest mortality (61.8%) but a lower percentage lost to follow-up (25.5%) compared to younger patients under 45 years, who had a lower mortality (15.6%) but a higher rate of loss to follow-up (37.8%). Unemployed patients showed higher

**Table 1. Sociodemographic characteristics of stroke patients followed for 12 months (n = 137).**

| Variable | Category | No. of patients | Survival Status | | p-value |
|---|---|---|---|---|---|
| | | | Deceased (n%) | Censored (n%) | |
| Sex | Female | 84 | 48(57.1) | 36(42.9) | 0.876 |
| | Male | 53 | 31(58.5) | 22(41.5) | |
| Hospital of Admission | Parirenyatwa Hospital | 47 | 33(70.2) | 14(29.8) | 0.005 |
| | Sally Mugabe Hospital | 54 | 22(40.7) | 32(59.3) | |
| | Chitungwiza Central Hospital | 36 | 24(66.7) | 12(33.3) | |
| Age Group (years) | < 45 | 28 | 7(25.0) | 21(75.0) | <0.001 |
| | 45–65 | 57 | 30(52.6) | 27(47.4) | |
| | > 65 | 52 | 42(80.8) | 10(19.2) | |
| Marital Status | Married | 79 | 46(58.2) | 33(41.8) | 0.876 |
| | Not in Union** | 58 | 33(56.9) | 25(43.1) | |
| Occupational Status | Unemployed | 75 | 46(61.3) | 29(38.7) | 0.017 |
| | Employed / self-employed | 36 | 14(38.9) | 22(61.1) | |
| | Retired | 26 | 19(73.1) | 7(26.9) | |
| Level of education | Primary | 60 | 47(78.3) | 13(21.7) | <0.001 |
| | Secondary | 65 | 24(36.9) | 41(63.1) | |
| | Tertiary | 9 | 5(55.6) | 4(44.4) | |
| Housing Status | Staying with parents / relatives | 36 | 17(47.2) | 19(52.8) | 0.216 |
| | Own House | 80 | 51(63.8) | 29(36.2) | |
| | Tenant | 21 | 11(52.4) | 10(47.6) | |
| Cigarette Smoking Status | Never smoked | 100 | 52(52.0) | 48(48.0) | 0.059 |
| | Former | 16 | 13(81.2) | 3(18.8) | |
| | Current smoker | 21 | 14(66.7) | 7(33.3) | |
| Hospital Bill Payment Modality | Free Service* | 22 | 13(59.1) | 9(40.9) | 0.239 |
| | Cash | 104 | 60(57.7) | 44(42.3) | |
| | Health insurance | 11 | 6(54.6) | 5(45.4) | |
| Transport Modality | Public | 117 | 66(56.4) | 51(43.6) | 0.472 |
| | Private | 20 | 13(65.0) | 7(35.0) | |
| Stroke Side | Bilateral | 1 | 0(0.0) | 1(100.0) | 0.241 |
| | Left | 68 | 36(52.9) | 32(47.1) | |
| | Right | 68 | 43(63.2) | 25(36.8) | |
| Comorbidities | No | 21 | 10(47.6) | 11(52.4) | 0.311 |
| | Yes | 116 | 69(59.5) | 47(40.5) | |
| Time of Stroke | Day | 93 | 53(57.0) | 40(43.0) | 0.816 |
| | Night | 44 | 26(59.1) | 18(40.9) | |
| Total Function | Mild/Moderate | 26 | 8(30.8) | 18(69.2) | <0.001 |
| | Severe | 27 | 10(37.0) | 17(63.0) | |
| | Very severe | 84 | 61(72.6) | 23(27.4) | |
| Dependency | Dependent | 131 | 77(58.8) | 54(41.2) | 0.217 |
| | Independent | 6 | 2(33.3) | 4(66.7) | |

Key: ** Not in Union includes, single, widowed, divorced and separated.

* 'Free Service' refers to social welfare, older than 65, pensioners / retirees

mortality (46.5%) but a lower percentage loss to follow-up (24.2%) compared to employed or self-employed patients, who had a lower mortality (24.2%) but a higher loss to follow-up rate (37.9%). Patients with primary level education only had the highest mortality (57.3%) with 26.8% loss to follow-up, while those with secondary level education had better survival

**Table 2. Sociodemographic characteristics of stroke patients including lost to follow-up/missing data (n = 188).**

| Variable | Category | No. of patients | Survival Status | | | p-value |
|---|---|---|---|---|---|---|
| | | | Deceased (%) | Censored (%) | Lost to follow-up/missing (%) | |
| Sex | Female | 119 | 48(40.3) | 36(30.3) | 35(29.4) | 0.644 |
| | Male | 69 | 31(44.9) | 22(31.9) | 16(23.2) | |
| Hospital of Admission | Parirenyatwa Hospital | 80 | 33(41.3) | 14(17.4) | 33(41.3) | p<0.001 |
| | Sally Mugabe Hospital | 60 | 22(36.7) | 32(53.3) | 6(10.0) | |
| | Chitungwiza Central Hospital | 48 | 24(50.0) | 12(25.0) | 12(25.0) | |
| Age Group (years) | < 45 | 45 | 7(15.6) | 21(46.6) | 17(37.8) | p<0.001 |
| | 45–65 | 75 | 30(40,0) | 27(36.0) | 18(24.0) | |
| | > 65 | 68 | 42(61.8) | 10(14.7) | 16(25.5) | |
| Marital Status | Married | 107 | 33(40.7) | 25(30.9) | 23(28.4) | 0.932 |
| | Not in Union** | 81 | 46(43.0) | 33(30.8) | 28(26.2) | |
| Occupational Status | Unemployed | 99 | 46(46.5) | 29(29.3) | 24(24.2) | 0.009 |
| | Employed / self-employed | 58 | 14(24.2) | 22(37.9) | 22(37.9) | |
| | Retired | 31 | 19(61.3) | 7(22.6) | 5(16.1) | |
| Level of education | Primary | 82 | 47(57.3) | 13(15.9) | 22(26.8) | p<0.001 |
| | Secondary | 86 | 24(27.9) | 41(47.7) | 21(24.4) | |
| | Tertiary | 15 | 5(33.3) | 4(26.7) | 6(40.0) | |
| Housing Status | Staying with parents / relatives | 52 | 17(32.7) | 19(36.5) | 16(30.8) | 0.359 |
| | Own House | 105 | 51(48.6) | 29(27.6) | 25(23.8) | |
| | Tenant | 31 | 11(35.4) | 10(32.3) | 10(32.3) | |
| Cigarette Smoking Status | Never smoked | 136 | 52(38.2) | 48(35.3) | 36(26.5) | 0.175 |
| | Former | 21 | 13(61.9) | 3(14.3) | 5(23.8) | |
| | Current smoker | 31 | 14(45.2) | 7(22.5) | 10(32.3) | |
| Hospital Bill Payment Modality | Free Service* | 27 | 13(48.2) | 9(33.3) | 5(18.5) | 0.675 |
| | Cash | 143 | 60(42.0) | 44(30.8) | 39(27.2) | |
| | Health insurance | 18 | 6(33.3) | 5(27.8) | 7(38.9) | |
| Transport Modality | Public | 161 | 66(41.0) | 51(31.7) | 44(27.3) | 0.761 |
| | Private | 27 | 13(48.2) | 7(25.9) | 7(25.9) | |
| Stroke Side | Bilateral | 2 | 0(0.0) | 1(50.0) | 1(50.0) | 0.065 |
| | Left | 83 | 36(43.4) | 32(38.6) | 15(18.1) | |
| | Right | 103 | 43(41.7) | 25(24.3) | 35(34.0) | |
| Comorbidities | No | 32 | 10(31.2) | 11(34.4) | 11(34.4) | 0.374 |
| | Yes | 156 | 69(44.2) | 47(30.1) | 40(25.7) | |
| Time of Stroke | Day | 127 | 53(41.7) | 40(31.5) | 34(26.8) | 0.961 |
| | Night | 61 | 26(42.6) | 18(29.5) | 17(27.9) | |
| Total Function | Mild/Moderate | 39 | 8(20.5) | 18(46.2) | 13(33.3) | p<0.001 |
| | Severe | 46 | 10(21.7) | 17(37.0) | 19(41.3) | |
| | Very severe | 103 | 61(59.2) | 23(22.3) | 19(18.5) | |
| Dependency | Dependent | 176 | 77(43.8) | 54(30.7) | 45(25.5) | 0.108 |
| | Independent | 12 | 2(16.7) | 4(33.3) | 6(50.0) | |

Key: ** Not in Union includes, single, widowed, divorced and separated.

* 'Free Service' refers to social welfare, older than 65, pensioners / retirees

outcomes (47.7% censored) and 24.4% loss to follow-up. In terms of stroke severity, very severe cases showed the highest mortality (59.2%) but lower loss to follow-up (18.5%) compared to those with mild/moderate strokes, who had lower mortality (20.5%) but higher loss to follow-up (33.3%).

## Overall Kaplan-Meier estimation of survival

The overall Kaplan-Meier estimation of survival, indicates that the overall probability of survival was above 50% during the first 100 days but declined to below 50% by day 200. Over the 12-month follow-up period from baseline, involving a total of 137 patients with stroke, 42% of individuals survived, while 58% were deceased [Fig 2].

## Kaplan-Meier survivor estimates of patients with stroke

Survival estimates for patients with stroke were analyzed using the Kaplan-Meier method to compare survival times. The results revealed distinct patterns based on functional outcomes. For instance, individuals with very severe functional outcomes at baseline had the shortest survival time, while those with mild and moderate outcomes survived the longest. Further comparison by age group demonstrated that patients below 45 years showed higher survival times, and the lowest survival times were observed in patients aged over 60 years. Additionally, based on participant housing status, patients staying with relatives or parents and those residing as tenants had longer survival times compared to those with own accommodation [Fig 3].

## AFT model selection

The efficiency of the AFT models was compared using the Akaike Information Criterion (AIC). Among the alternative AFT models considered (Weibull, Exponential, Log-normal), the Gompertz AFT model showed the lowest AIC value (AIC = 395.2), signifying its best fit to the stroke dataset compared to other alternatives. Consequently, the Gompertz AFT model was chosen as the most suitable model for describing the characteristics of the stroke dataset (Table 3).

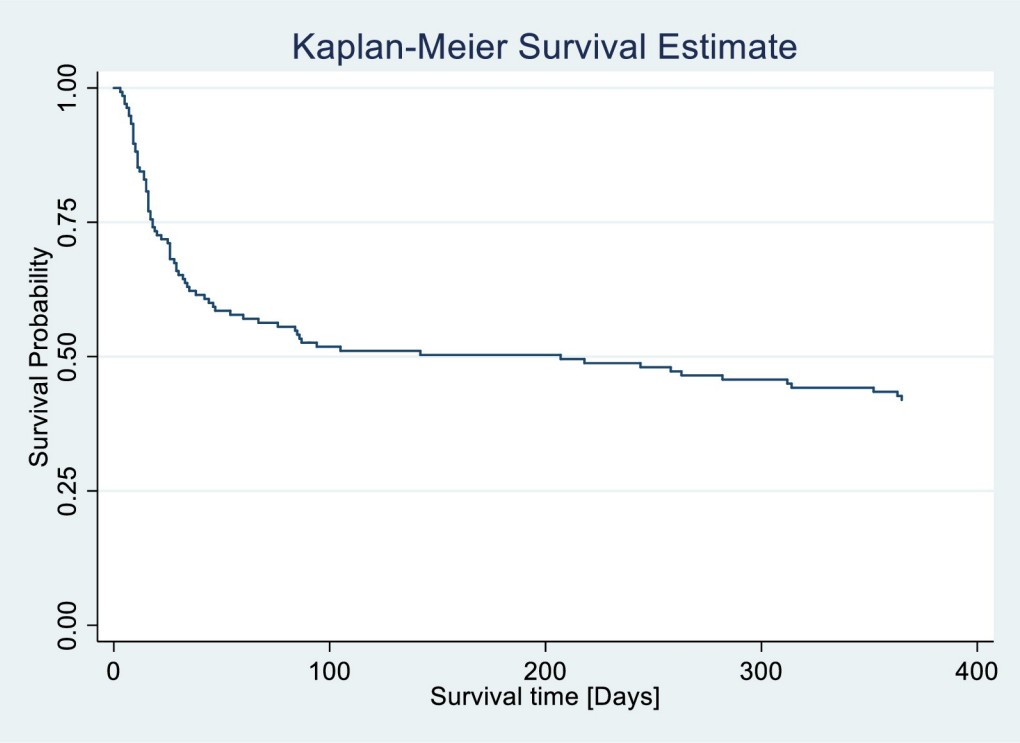

**Fig 2. Overall Kaplan–Meier survival status among stroke patients.**

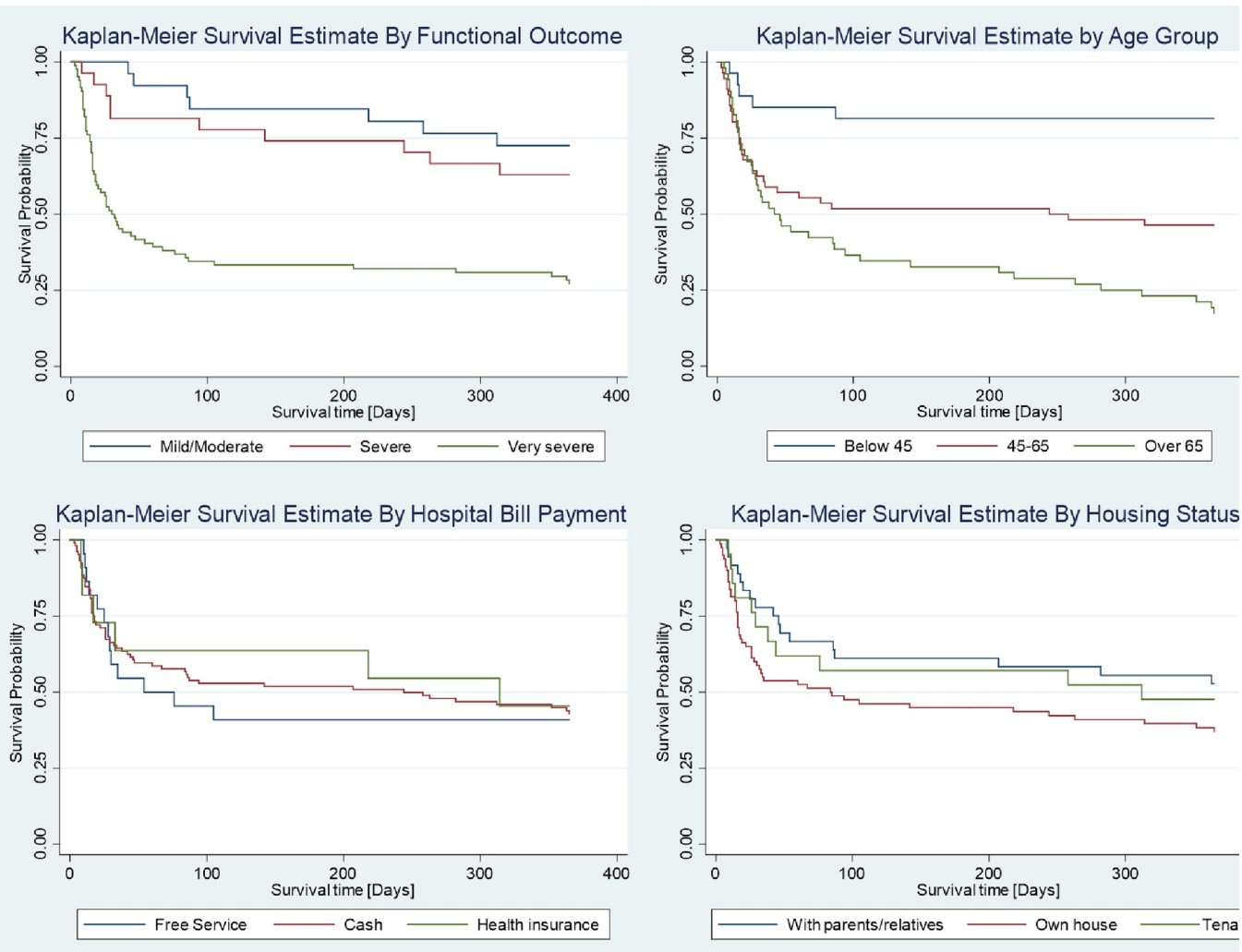

**Fig 3. Kaplan-Meier survivor estimates by functional outcome, age group, hospital bill payment and housing status.**

## AFT models graphical evaluation

The adequacy of model fit was further evaluated through graphical analysis using Cox-Snell residuals. Specifically, Cox-Snell residuals were employed to assess the overall goodness of fit across the different parametric models.

Fig 4 illustrates the Cox-Snell residuals plots, revealing that the Gompertz AFT model demonstrated the most favorable fit for this dataset of stroke patients. This determination was

**Table 3. Akaike's information criterion values for parametric models.**

| Model | AIC |
|---|---|
| Weibull | 416.0 |
| Exponential | 433.7 |
| Gompertz | 395.2 |
| Log-Normal | 400.7 |

**Key:** AIC-Akaike information criterion

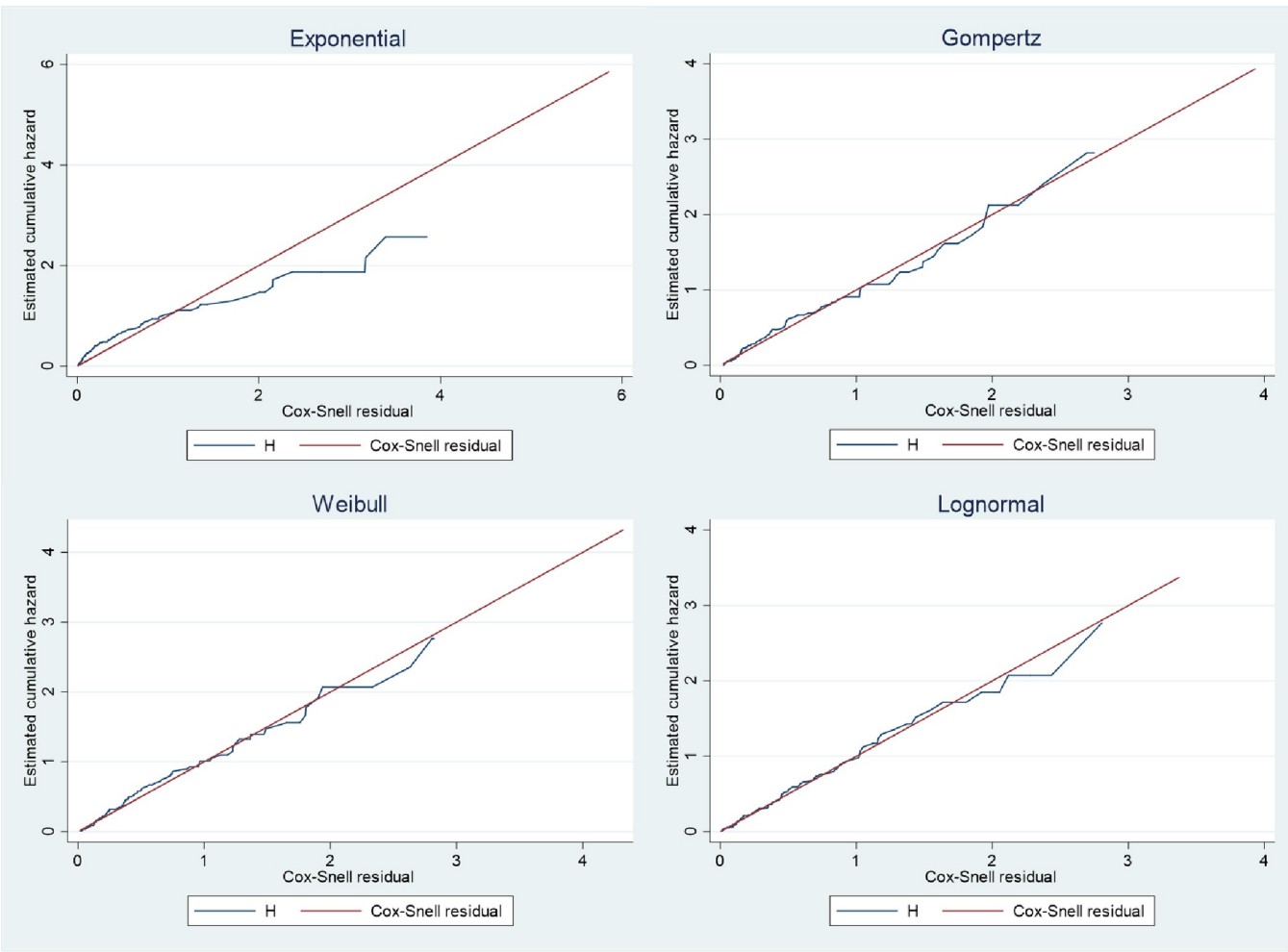

**Fig 4. Estimated cumulative hazard against Cox-Snell residuals for the Exponential, Gompertz, Weibull and Lognormal models.**

based on the observation that the plot of Cox-Snell residuals against the cumulative hazard function of residuals formed an approximately straight line through the origin at a 45-degree angle, distinguishing it as the optimal fit when compared to the Exponential, Weibull, and lognormal models.

## Factors influencing survival among stroke patients

In the multivariable Gompertz AFT model for survival status among stroke patients (Table 4), we included factors that were deemed eligible for further analysis based on the bivariate analysis. Regarding hospital bill payment modalities, compared to cash payments, those receiving free services which included social welfare, older than 65, pensioners and retirees had a significantly lower risk of death 0.4(95%CI:0.20, 0.80), implying a protective effect on survival. However, health insurance modality was not significantly associated with survival. Educational attainment demonstrated a significant trend, with higher educational attainment associated with lower risk of mortality. For instance, individuals with secondary school level education had adjusted hazards ratios of 0.4(95% CI: 0.24, 0.79), indicating a significantly reduced risk compared to those with primary education. Age was also significantly associated with mortality, with individuals aged 45-65years and those aged over 65 years showing adjusted hazards

**Table 4. Multivariable Gompertz AFT model for survival status and predictors among stroke patients (n = 137).**

| Variable | Category | Adjusted HR 95% CI | P > \| z \| |
|---|---|---|---|
| Hospital Bill Payment modality | Cash | Reference | |
| | Free service | 0.4 (0.20, 0.80)** | 0.009 |
| | Health insurance | 0.6 (0.27, 1.57) | 0.336 |
| Level of Education | Primary | Reference | |
| | Secondary | 0.4 (0.24, 0.79)** | 0.007 |
| | Tertiary | 0.6 (0.24, 1.73) | 0.386 |
| Age Group | < 45 | Reference | |
| | 45–65 | 3.4 (1.42, 8.36)** | 0.006 |
| | > 65 | 3.7 (1.44, 9.36)** | 0.006 |
| Housing Status | Own House | Reference | |
| | Staying with parents / relatives | 0.4 (0.20, 0.64)*** | 0.001 |
| | Tenant | 0.9 (0.46, 1.85) | 0.825 |
| Total Function | Very severe | Reference | |
| | Mild / Moderate | 0.2 (0.09, 0.40)*** | <0.001 |
| | Severe | 0.2 (0.10, 0.46)*** | <0.001 |

**Key:** * p<0.05

**p<0.01

***p<0.001

ratios of 3.4(95%CI:1.42, 8.36) and 3.7(95%CI:1.44, 9.36) respectively, compared to those below 45 years. Housing status showed a significantly protective effect for those staying with parents/relatives [0.4(95%CI: 0.20, 0.64)]. Total functional outcome showed significantly lower hazard for individuals with mild or moderate and severe outcomes compared to those with very severe outcomes, with hazard ratios of 0.2(95%CI: 0.09, 0.40) and 0.2 (95%CI: 0.10, 0.46), respectively.

## Discussion

This longitudinal study enrolled 188 stroke patients and sought to uncover influential determinants that influenced survival outcomes among stroke patients over a 12-month period. Our results indicate that the overall probability of survival was above 50% during the first 100 days but declined to below 50% by day 200. Overall, 79 (58%) stroke patients died during the 12 months' period. This seemingly high proportion of deaths could be attributed to complications that develop among stroke patients over time as well as complications of bed rest. These results showed higher mortality at 3 months when compared to the pooled case fatality for SSA which was reported to be 22.3% [4]. Our finding is however, in concordance with global trends as between 20 to 75% per cent of first-time patients with stroke reportedly die within a month, and a third by 6 months [22, 23]. Overall case fatality rates (CFR) have been reported to range from 18% to 62% in some African countries [4, 24, 25]. Pooled CFR for SSA at one year was reported to be 33.4% [4] which figure is also lower than our results. Contrasting results showing much lower death rates of 2.5% in the first year of follow-up were reported in Thailand [22]. Furthermore, CFR of 20% were reported in other countries of the world with CFR rates of 4% reported from high income countries [4] The differences could potentially stem from variations in study design, including differences in patient demographics, healthcare systems, treatment protocols, and follow-up durations.

Although varied, the mortality rates in different African countries could be due to limited health care facilities and uncontrolled risk factors, poor functional abilities post-stroke, older age and impaired consciousness [26–29]. Another explanation could be that most patients could not afford to go to health centres for prescription refills due to a lack of finances before stroke. Thus, once such patients suffered a stroke, their conditions worsened, further compounded by inability to travel or fund medications [29, 30]. In addition, only one hospital among the three hospitals in Zimbabwe had a stroke unit at the time of the study whilst the other two hospitals had neurologists managing the patients. In other countries, progress in stroke care has been made with the establishment of stroke units leading to a significant reduction in the one- month stroke mortality [4, 30].

Findings from the present study, provide valuable insights into the factors influencing survival outcomes among stroke patients. We previously reported an in-hospital fatality rate of 25% in a one-year retrospective study at these three same hospitals [31]. This disparity with the current study (58%) could be attributed to differences in study populations, such as variations in demographics, co-morbidities and stroke severity. These may have contributed to the higher fatality rate observed in the current study. In addition, the changes in healthcare access and quality of care over time might have impacted patient outcomes. Furthermore, the retrospective study included only in-hospital deaths. The multi-variable Gompertz AFT model allowed for a comprehensive assessment of various predictors, shedding light on the influence of socio-demographic and clinical variables on survival time of individuals' post-stroke.

Educational attainment emerged as a significant predictor of mortality post stroke, unveiling a trend where a reduced risk of death was observed among those with higher educational attainment. Particularly noteworthy was the significantly lower risk observed in individuals with secondary school level education compared to those with primary school level education. This finding was in concordance with previous research findings [10, 11, 30]. This highlights the pivotal role of education as a social determinant of health, influencing factors such as health literacy, access to resources, and possibly adherence to medical recommendations. Interestingly, the better outcomes associated with secondary level education, compared to both primary and tertiary level education, may be linked to socio-economic and lifestyle factors. Individuals with secondary level education likely occupy a middle socio-economic tier, with better access to healthcare and healthier lifestyles. In contrast, those with tertiary level education may face higher levels of occupational stress, contributing to risk factors like hypertension or sedentary behaviour, while individuals with primary level education may lack health literacy or access to adequate healthcare, leading to poorer stroke outcomes. This pattern reflects the complex ways in which education interacts with socio-economic status and health behavior to influence health outcomes [32]. Understanding these dynamics is critical, particularly in low-resource settings where disparities in education and healthcare access can significantly affect disease outcomes.

Age was also an influential determinant of mortality within the one-year follow-up period. Individuals aged 45–65 and over 65years showed substantially higher hazard ratios compared to those below 45 years. This also aligns with existing studies [6, 7, 33], emphasizing the impact of age on stroke outcomes and reinforces the important need for tailored care strategies that consider the unique needs and challenges associated with different age groups. Tailoring interventions based on age-specific considerations may enhance the effectiveness of stroke care and improve overall survival rates.

Housing status emerged as an important determinant of stroke outcomes, with a protective effect observed for individuals living with parents or relatives. This suggests that residing with family members may create a supportive environment that mitigates the risk of mortality, highlighting the potential influence of social support systems beyond housing status alone.

The improved outcomes associated with living with family may also be influenced by factors such as younger age and fewer comorbidities among such survivors, making them more likely to recover from stroke. Furthermore, family support plays a crucial role in enhancing health outcomes by facilitating timely access to care, promoting adherence to treatment regimens, and fostering more effective overall health management. In contrast, patients living independently may delay seeking treatment, contributing to poorer outcomes. While our analysis identified this association, it is important to acknowledge the role that age and stroke severity play in shaping these results. Future research should explore the impact of family support on stroke recovery, especially in resource-limited settings where informal caregiving is a critical component of patient care.

Total functional outcome, a key aspect of stroke management, surfaced as a significant predictor in this study. Individuals classified as having 'Very severe' stroke showed an increased hazard of death. This demonstrates the critical importance of assessing and addressing functional outcomes in stroke management, as severe functional impairment significantly amplifies the risk of mortality. Tailoring interventions to address functional limitations and promote rehabilitation may contribute to improved survival outcomes for stroke patients. Previous studies have also highlighted an association between functional impairment, specifically cognitive impairment, and an elevated risk of mortality among individuals affected by stroke [34].

The observed differences in survival times across hospital bill payment modalities might reflect not only economic disparities but also differences in health-seeking behaviors and illness severity at the time of hospital admission. Non-paying patients, comprising social welfare recipients, pensioners, retirees, and individuals aged 65+ years, are likely to seek healthcare services early. This enables them to seek care for less severe conditions potentially contributing to their improved survival outcomes. Conversely, cash paying patients may delay seeking care due to financial barriers, only presenting at hospitals when their condition becomes critical. This difference in access to timely care possibly contributes to the poorer outcomes seen in the privately paying group. An earlier study showed that cash paying patients present later to hospital as they may not have money for hospital services and transport, hence poorer survival outcomes as their conditions may deteriorate while waiting for financial support [35]. This demonstrates the impact of economic barriers on health outcomes, particularly in contexts like Zimbabwe, where financial constraints may limit access to healthcare. The relationship between healthcare payment modalities and survival outcomes warrants further investigation, as it reflects broader socio-economic inequalities that can significantly influence recovery and long-term survival. It also points towards the need for research on patients and caregiver health seeking behaviors as results may imply that some of the paying survivors may have to wait for decisions made by caregivers who may answerable to another group of caregivers within the hierarchical families.

While the Gompertz AFT model identified significant associations between the observed socio-economic and demographic factors and stroke outcomes, it is essential to emphasize that these associations do not imply direct causation. For example, the observed better outcomes for survivors living with parents may be linked to their younger age or less severe strokes, rather than the living arrangement itself being a direct cause of improved survival. The model was selected based on its superior fit for the data compared to other models. However, like any statistical model, it identifies relationships that need to be considered in the broader clinical and social context. The intent behind employing the Gompertz AFT model was to provide a more precise analysis of time-to-event data, rather than to establish definitive cause-and-effect links.

## Strengths and limitations of the study and or findings

The findings of the present study highlighted the factors associated with mortality in this group of patients and paves way for health professionals to develop guidelines to mitigate mortality among stroke patients. The limitations of this study include missing information about the cause of death which might have introduced potential selection bias despite efforts to minimize it. This led to analysis only for the available data with a substantially diminished sample size thus statistical analysis on causality could not be carried out. Our study was only on patients who were admitted in the hospitals and did not include those who died before admission and those who were not admitted due to various reasons. Wide confidence intervals on some variables suggest a possible need for replication of the study with a much larger sample size. The 12-month follow-up might have limited capturing long-term survival patterns, potentially missing late-stage events and ongoing health challenges beyond this time frame. In addition, the exclusion of analysis by HIV status, due to the large proportion of participants with missing HIV status data, is a limitation of this study. Future studies should focus on collecting more complete HIV data and analyse its potential impact on stroke outcomes, as HIV infection may significantly influence patient prognosis and recovery.

## Conclusions

The observed influence of hospital bill payment modalities on survival outcomes, with receiving free hospital services being associated with a lower risk of death, suggests potential avenues for healthcare policy changes and practice improvements. Additionally, the significant trend observed in educational level attainment underscores its possible role as a crucial social determinant of health in the context of stroke since educational attainment influences health literacy, resource access, and adherence to medical recommendations. The critical impact of age on survival outcomes emphasizes the need for tailored care strategies for different age groups. Furthermore, the protective effect of staying with parents/relatives carers and the heightened mortality risk associated with 'Very severe' functional outcomes emphasize the importance of assessing and addressing social support systems and functional impairment in stroke management. This highlights the nature of stroke prognosis, emphasizing the need for patient-centered approach in stroke care that considers and addresses diverse determinants impacting survival outcomes. Community based studies may give more information about case fatality rates among stroke patients in Zimbabwe.

## Supporting information

**S1 File. The inclusivity in global research outlines the ethical, cultural, and scientific considerations specific to inclusivity in global research that we carried out during our study.** (PDF)

**S1 Data set. This is the data set that we used to reach the conclusions drawn in the manuscript with related metadata and methods.** This data can be replicated to report the study findings in their entiretyincluding a.) The values behind the means, standard deviations and other measures reported; b.) The values used to build graphs; c.) The points extracted from images for analysis. (XLSX)

## Acknowledgments

Firstly, we would like to acknowledge the participants of the study. Our special thanks and appreciation also goes to the research assistants and staff of the three public referral hospitals; Parirenyatwa, Sally Mugabe and Chitungwiza Central Hospitals in Zimbabwe.

## Author Contributions

**Conceptualization:** Farayi Kaseke, Lovemore Gwanzura, Aimee Stewart.

**Data curation:** Farayi Kaseke.

**Formal analysis:** Cuthbert Musarurwa, Tawanda Nyengerai.

**Funding acquisition:** Farayi Kaseke.

**Investigation:** Farayi Kaseke, Lovemore Gwanzura, Timothy Kaseke.

**Methodology:** Farayi Kaseke, Lovemore Gwanzura.

**Project administration:** Farayi Kaseke, Timothy Kaseke.

**Resources:** Farayi Kaseke, Timothy Kaseke.

**Software:** Cuthbert Musarurwa, Tawanda Nyengerai.

**Supervision:** Lovemore Gwanzura, Aimee Stewart.

**Validation:** Elizabeth Gori, Tawanda Nyengerai.

**Visualization:** Tawanda Nyengerai.

**Writing – original draft:** Farayi Kaseke, Cuthbert Musarurwa, Elizabeth Gori, Tawanda Nyengerai, Timothy Kaseke.

**Writing – review & editing:** Farayi Kaseke, Lovemore Gwanzura, Cuthbert Musarurwa, Elizabeth Gori, Tawanda Nyengerai, Timothy Kaseke, Aimee Stewart.

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
