## [Decision Letter · Decision Letter 0]

9 Jul 2024

PONE-D-24-12258Factors influencing survival outcomes in patients with stroke in Zimbabwe: A 12-month longitudinal studyPLOS ONE

Dear Dr. Kaseke,

Thank you for submitting your manuscript to PLOS ONE. After careful consideration, we feel that it has merit but does not fully meet PLOS ONE’s publication criteria as it currently stands. Therefore, we invite you to submit a revised version of the manuscript that addresses the points raised during the review process.

 Please carefully review the comments provided by the reviewers, particularly points raised regarding the inclusion of additional details. 

We look forward to receiving your revised manuscript.

Kind regards,

Ryan G Wagner, MSc(Med), MBBCh, PhD

Academic Editor

PLOS ONE

“Firstly, we would like to acknowledge the participants of the study. Our special thanks and appreciation also goes to the research assistants and staff of the three public referral hospitals; Parirenyatwa, Sally Mugabe and Chitungwiza Central Hospitals in Zimbabwe. We would also want to acknowledge the funders of the study - NIH Fogarty student support grant (Grant Number D43TW009539).”

“Farayi Kaseke was supported by the Training for Research Excellence and Mentorship in Tuberculosis (TRENT) Program, Grant Number D43TW009539. from the National Institute of Health (NIH) as a PhD scholar. This grant was awarded in collaboration with the University of Zimbabwe  Faculty of Medicine. The funding was used for data collection purposes. The funding grant ended on 31st December 2023.”

4. In the online submission form, you indicated that [Data is available upon request.].

Reviewers' comments:

Reviewer's Responses to Questions

**Comments to the Author**

1. Is the manuscript technically sound, and do the data support the conclusions?

Reviewer #1: Yes

Reviewer #2: Partly

Reviewer #3: Yes

2. Has the statistical analysis been performed appropriately and rigorously? 

Reviewer #1: Yes

Reviewer #2: No

Reviewer #3: No

3. Have the authors made all data underlying the findings in their manuscript fully available?

Reviewer #1: Yes

Reviewer #2: Yes

Reviewer #3: Yes

4. Is the manuscript presented in an intelligible fashion and written in standard English?

Reviewer #1: Yes

Reviewer #2: No

Reviewer #3: Yes

5. Review Comments to the Author

Reviewer #1: Overall

Introduction

The background provides a clear rationale for the study. It might benefit from expanding on why these particular determinants were added (Line 72-73).

Methods

The study design, data collection, and analysis methods are well-documented. The manuscript addresses the failure of the Cox proportional hazards model and appropriately shifts to the AFT model, which is well-justified. However, it would be helpful to include more information on the validation of the data collection tools and potential sources of bias.

Results

The results are presented clearly, with appropriate use of tables and figures. The Kaplan-Meier survival curves and Cox-Snell residuals plots effectively illustrate the findings.

Discussion

The discussion appropriately interprets the results, relating them to existing literature. The limitations section is honest and acknowledges the potential biases and the limited sample size. Future studies could aim for a larger sample size and a longer follow-up period to capture more long-term survival patterns.

Specific

• Heading - The heading could benefit from being more specific. Consider including that this study was conducted in three tertiary hospitals to provide clearer context.

• Abstract - The first hazard ratio (HR) mentioned in the abstract may need to be written out in full, and then subsequent hazard ratios can be abbreviated to maintain clarity.

• Sample Size - There was a 27% loss in the sample size. It would be helpful to know if this was accounted for in the initial sample size calculation and its potential impact on the study's statistical power.

• Exclusion Criteria (Line 97-96) - Providing a rationale for excluding Transient Ischaemic Attacks (TIA) would strengthen the clarity and justification of the exclusion criteria.

• Operational Definitions - Including a definition of stroke would be beneficial. Was the study looking at all types of Stroke or just Ischaemic Strokes?

• Data Collection Process - Was the principal investigator present to conduct data quality checks upon collection by the research assistants? Clarifying this would ensure confidence in data integrity.

• Variables

- Under hospital bill modalities, how does 'Free Service' differ from 'Social Welfare'?

- How was the total function of the patients measured? Provide more detail on this measurement to enhance understanding.

• Results Table - The results table could benefit from including specific p-values for the important variables to give readers a clearer sense of statistical significance.

• Discussion (Line 60-62) - The Thailand study mentions contrasting results, what are the Author’s thoughts on this. Could these differences be due to variations in study design? Expanding on this shows critical evaluation of the paper.

• Discussion (Line 67-68) - There are minor spelling errors ("worsen" and "further") that need correction.

• Discussion (Line 77) - The statement that higher educational attainment "correlated" with a reduced risk of death needs to be substantiated. The findings indicate that secondary education had a higher survival rate, but the results for primary and tertiary education were not significant in the Gompertz AFT model. Survival status in Table 1 also shows a higher survival status in those with secondary compared to tertiary education. Thus, perhaps reconsideration of the word ‘correlated’ is needed.

• Discussion (Line 92-96) - The impact of housing status could be confounded by the presence of a support system. The conclusion stated is about support system, however the data collected is on housing status alone. Clarify whether the study is assessing the effect of having one's own house or the presence of relatives in the house.

• Rehabilitation Attendance - The study could benefit from assessing trends in rehabilitation attendance post-discharge. If data on this is available, reporting it would add depth to the findings.

• Hospital Payment Modalities - What were the specific p-values for health insurance and social welfare? If these values are between 0.05 and 0.1, it could strengthen your argument that these variables warrant further exploration.

• Strengths and Limitations - The statement that recruitment from central hospitals improves generalisability may need reconsideration. Patients at central hospitals are often referred for advanced care, which may not represent the broader population.

Overall, the manuscript is a valuable contribution to understanding stroke survival outcomes in low-resource settings. The findings have important implications for healthcare policy and stroke management in Zimbabwe and similar contexts

Reviewer #2: Editorial/English language revision required; typos, sentencing and the like needs to be thoroughly revised; please may respect be payed to grammar and spelling. A manuscript of this prospected calibre should be devoid of major errors in language (page 3, line 63/64: "necessitate the need" - this is atrocious English...).

The study design:

It appears that this paper is a sub-analysis of a "larger" (page 4, 86) study which I don't see referred to in reference or text; was the data collected prospectively? it appears so, but is not explicitly stated. Who identified the patients? How were they recruited? This is crucial to the understanding of the message.

Do the respective hospitals have stroke units? The three hospitals were manned by neurologists? (page 4,89). I am not aware that this is the case, never mind weekly outpatient stroke clinics for "all" stroke patients discharged...

Page 6 and page 8, the Ethics statement is duplicated.

Results:

Page 8:

The gross discrepancy in mortality between the three hospitals needs to be explored and further explained: is it different populations with different pathologies, different referral pathways or else? Are these all tertiary referral centres, if so why these stark contrasts?

Page 9:

86: regarding level of education, only three (3) patients were uneducated (no formal education), all of whom died within the 12 months follow up, obviously making it 100%; this is too small a number to even mention!!!

Page 10:

Comorbidities: saying that 21 patients had no comorbidities; this is from a clinician's point of view very unlikely and hard to believe.

Was HIV status known in the cohort or undetermined or ignored? This would have been very valuable info.

The outcomes/survival times according to payer categories don't make sense as the non-paying ones are better off (but these are older pensioners etc.). The privately paying patients had poorer survival, but this may be related to the fact that these were sicker patients with no choice but biting the "private" bill? People in this category will not go to hospital for minor strokes. Patients staying with parents/relatives have better survival because they are probably younger...

Here is the crux of the paper and my main criticism which can be allayed by proper/different statistics being applied:

The authors seem to find associations by the Gompertz AFT modelling statistics because Cox bivariate regression had failed to show significant correlations between the variables and time outcomes. The authors derive correlations based on the AFT stats, but seem to mix up association and causation: the fact that people living with parents/relatives have better outcomes than people living in their own homes may be related to the fact that the former are younger than the latter... These patients have a better outcomes because their strokes might be milder and they can "afford" to seek hospitalisation for minor strokes as opposed to others.

In my view this data set should have been processed by a multivariate regression analysis taking into account the multitude of variables involved in this complex setting of societal and economic factors. The authors make too little and simple reading of what the AFT calculations appear to suggest: there are far too many contradictory outcomes explained too simply as cause and effect.

Reviewer #3: Major comments

1. Table 1 does not show p-values of the comparisons between those who survived stroke versus those deceased.

2. There is substantial loss to follow-up. Please kindly provide a Table that compares those who were lost to follow-up versus those who stayed in the study for analysis.

3. Discussion: A sample size of 137 is not a ‘large sample size’ to be reported as a strength of the study. Please remove this statement.

4. Discussion: The authors should please discuss the reasons for the disparity in stroke fatality of 25% in their previous study juxtaposed against 58% in the current study.

5. Adoukounou et al has reported a meta-analysis of stroke fatality in Africa of 33.2%. Authors should please discuss their findings in the light of this meta-analytic data.

6. The use of reference [1] is not correct as this is a Review article. The incidence, prevalence, mortality rates of stroke in Africa have specific studies which should be cited, not the Review article which summarizes these findings.

6. PLOS authors have the option to publish the peer review history of their article (what does this mean?). If published, this will include your full peer review and any attached files.

Reviewer #1: No

Reviewer #2: **Yes: **Andre Mochan

Reviewer #3: **Yes: **FRED STEPHEN SARFO

---

## [Author Response · Author response to Decision Letter 0]

12 Aug 2024

REVIEWER COMMENTS 

1. Please ensure that your manuscript meets PLOS ONE's style requirements, including those for file naming. The PLOS ONE style templates can be found at;

 Response: Done

2. Please include a complete copy of PLOS’ questionnaire on inclusivity in global research in your revised manuscript.

Please find more information on the policy and a link to download a blank copy of the questionnaire here: https://journals.plos.org/plosone/s/best-practices-in-research-reporting. Please upload a completed version of your questionnaire as Supporting Information when you resubmit your manuscript. 

Response: We included it

3. We note that you have provided funding information that is currently declared in your Funding Statement. However, funding information should not appear in the Acknowledgments section or other areas of your manuscript. We will only publish funding information present in the Funding Statement section of the online submission form. Please remove any funding-related text from the manuscript and let us know how you would like to update your Funding Statement. 

Currently, your Funding Statement reads as follows:

“Farayi Kaseke was supported by the Training for Research Excellence and Mentorship in Tuberculosis (TRENT) Program, Grant Number D43TW009539. from the National Institute of Health (NIH) as a PhD scholar. This grant was awarded in collaboration with the University of Zimbabwe Faculty of Medicine. The funding was used for data collection purposes. The funding grant ended on 31st December 2023.”

Response: We deleted the funding information from the Acknowledgements section and included it in the cover letter

4. In the online submission form, you indicated that [Data is available upon request.].

Response: Data was uploaded as a supplementary file (Excel file)

Response: This was done

Reviewer #1: Overall 

Introduction

The background provides a clear rationale for the study. It might benefit from expanding on why these particular determinants were added (Line 72-73). 

Response: This was added: (Line 82-87). Hospital bill payment modalities can reveal economic barriers and financial stress that affect recovery trajectories post-treatment(1,2). Educational attainment may influence health literacy, access to healthcare resources, and ultimately, health outcomes(3,4). Housing status is linked to social support levels and environmental stability, both of which are pivotal for long-term recovery and well-being(5). Total functional outcomes provide a measure of recovery, encompassing physical, cognitive, and social dimensions(6). This study sought …

Methods

However, it would be helpful to include more information on the validation of the data collection tools and potential sources of bias. 

Response: This was added: (Line 158-159). The data collection tool was validated for reliability before deployment. The Functional Independence Measure (FIM™) also used in this study, was validated and found to be a reliable tool for stroke patients(7–9).

Results and Discussion No issues raised

Specific

Heading - The heading could benefit from being more specific. Consider including that this study was conducted in three tertiary hospitals to provide clearer context. 

Response: Topic rephrased to: Factors influencing survival outcomes in patients with stroke at three tertiary hospitals in Zimbabwe: A 12-month longitudinal study

Abstract - The first hazard ratio (HR) mentioned in the abstract may need to be written out in full, and then subsequent hazard ratios can be abbreviated to maintain clarity. 

Response: This was corrected: (Line 46). (Hazard Ratio; HR: 0.4, 95% CI: 0.20, 0.80).

Sample Size - There was a 27% loss in the sample size. It would be helpful to know if this was accounted for in the initial sample size calculation and its potential impact on the study's statistical power. 

Response: This was added: Line 102-105). The initial sample size calculation accounted for anticipated attrition rates, to maintain the study's statistical power despite the loss. This consideration was to ensure that the conclusions drawn were based on a more homogeneous and complete dataset. However, attrition was higher than anticipated hence a limitation of the study

Exclusion Criteria (Line 97-96) - Providing a rationale for excluding Transient Ischaemic Attacks (TIA) would strengthen the clarity and justification of the exclusion criteria. 

Response: This was rephrased: Line 119-121). The study excluded admitted patients who died before medical stabilization or refused to participate, as well as those who had a Transient Ischaemic Attack (TIA). The rationale for excluding TIA cases was to ensure the study population consisted of patients with more definitive and severe cerebrovascular events, thereby reducing variability and improving the precision of the study's outcomes.

Operational Definitions - Including a definition of stroke would be beneficial. Was the study looking at all types of Stroke or just Ischaemic Strokes? 

Response: This was added under operational definitions (Line 139-151). Stroke: Stroke was defined as a focal neurological deficit caused by a disturbance of the circulation to the brain following an acute or sudden disturbance of brain function vascular in origin causing disability lasting more than, or death within 24 hours. The study included both ischaemic and hemorrhagic strokes.

Data Collection Process - Was the principal investigator present to conduct data quality checks upon collection by the research assistants? Clarifying this would ensure confidence in data integrity. 

Response: This was added: (Line 156-157). The principal investigator was present to conduct data quality checks upon collection by the research assistants.

Variables

- Under hospital bill modalities, how does 'Free Service' differ from 'Social Welfare'?

- How was the total function of the patients measured? Provide more detail on this measurement to enhance understanding. 

Response: "Social welfare" was combined with "free service” since both categories represent services provided at no direct cost to individuals. This will reflect 16.06% of the population benefiting from the support mechanisms. Appropriate changes were made in the manuscript.

This was further added to the FIM definition under data collection procedure: Total score for FIM was measured as the sum of the motor and cognition subscale scores. 

Results Table - The results table could benefit from including specific p-values for the important variables to give readers a clearer sense of statistical significance. 

Response: P-values were added on Table 1 and 3. 

Discussion (Line 60-62) - The Thailand study mentions contrasting results, what are the Author’s thoughts on this. Could these differences be due to variations in study design? Expanding on this shows critical evaluation of the paper. 

Response: This was added in discussion section: The differences could potentially stem from variations in study design, including differences in patient demographics, healthcare systems, treatment protocols, and follow-up durations. 

Discussion (Line 67-68) - There are minor spelling errors ("worsen" and "further") that need correction. Original: Thus once they suffered a stroke, their conditions got worsen being furthers compounded by inability to travel or fund medications

Response: Revised, thus once they suffered a stroke, their conditions worsened, further compounded by the inability to travel or fund medications : 94-95

Discussion (Line 77) - The statement that higher educational attainment "correlated" with a reduced risk of death needs to be substantiated. The findings indicate that secondary education had a higher survival rate, but the results for primary and tertiary education were not significant in the Gompertz AFT model. Survival status in Table 1 also shows a higher survival status in those with secondary compared to tertiary education. Thus, perhaps reconsideration of the word ‘correlated’ is needed. 

Response: Revised, educational attainment emerged as a significant predictor of mortality post stroke, unveiling a trend where a reduced risk of death was observed among those with higher education. Particularly noteworthy was the significantly lower risk observed in individuals with secondary school education compared to those with primary education….

Discussion (Line 92-96) - The impact of housing status could be confounded by the presence of a support system. The conclusion stated is about support system, however the data collected is on housing status alone. Clarify whether the study is assessing the effect of having one's own house or the presence of relatives in the house. 

Response: Revised, housing status emerged as another determinant, revealing a protective effect for those living with parents/relatives. This suggests that the presence of relatives in the house may offer a supportive environment that mitigates mortality risk. This implies that the observed protective effect could be influenced by the support system inherent in living with family, rather than housing status alone.

Rehabilitation Attendance - The study could benefit from assessing trends in rehabilitation attendance post-discharge. If data on this is available, reporting it would add depth to the findings. 

Response: All the patients were presumed to have been receiving the usual rehabilitation care but unfortunately, data on rate of rehabilitation attendance post-discharge were not available for analysis in this study. 

Hospital Payment Modalities - What were the specific p-values for health insurance and social welfare? If these values are between 0.05 and 0.1, it could strengthen your argument that these variables warrant further exploration. 

Response: The p-values were incorporated into the analysis as recommended. "Social Welfare" was combined with "Free Service" as both categories represent services provided at no direct cost to individuals, collectively reflecting 16.06% of the population benefiting from these support mechanisms. This consolidation was to simplify analysis and enhance the clarity of the impact of cost-free services.

Strengths and Limitations - The statement that recruitment from central hospitals improves generalisability may need reconsideration. Patients at central hospitals are often referred for advanced care, which may not represent the broader population.

Response: This statement was removed.

Reviewer #2 

Editorial/English language revision required; typos, sentencing and the like needs to be thoroughly revised; please may respect be payed to grammar and spelling. A manuscript of this prospected calibre should be devoid of major errors in language (page 3, line 63/64: "necessitate the need" - this is atrocious English...). 

Response: Revised: Understanding factors influencing survival outcomes, will provide a foundation for targeted interventions customized to the region’s unique socio-economic and healthcare landscape.

The English language was revised 

The study design:

It appears that this paper is a sub-analysis of a "larger" (page 4, 86) study which I don't see referred to in reference or text; was the data collected prospectively? it appears so, but is not explicitly stated. Who identified the patients? How were they recruited? This is crucial to the understanding of the message.

Do the respective hospitals have stroke units? The three hospitals were manned by neurologists? (page 4,89). I am not aware that this is the case, never mind weekly outpatient stroke clinics for "all" stroke patients discharged... 

Response: Revised: This paper is a sub-analysis of a larger study carried out to develop, implement and preliminarily evaluate outcomes of stroke survivors and their caregivers in Harare (18). The data were collected prospectively. The first author and research assistants identified the patients based on the inclusion criteria and consecutive sampling was done with patients recruited as they were admitted from each of the three hospitals.

At the time of data collection, only Parirenyatwa Group of Hospitals had a stroke unit. Parirenyatwa and Harare Central Hospitals were manned by neurologists while Chitungwiza hospital was manned by general practitioners and physicians who were not specifically neurologists. Although weekly outpatient stroke clinics for "all" stroke patients discharged are expected at these hospitals, it was no longer possible due to reduced numbers of physiotherapists and occupational therapists manning the out-patient clinics hence patients were seen once or twice a month.

Page 6 and page 8, the Ethics statement is duplicated. 

Response: Ethics under data collection procedures was removed, and ethics details were retained under the section for 'Ethics approval and participant consent'.

Results:

Page 8:

The gross discrepancy in mortality between the three hospitals needs to be explored and further explained: is it different populations with different pathologies, different referral pathways or else? Are these all tertiary referral centres, if so why these stark contrasts? 

Responses: This likely arises from differences in referral pathways and hospital capabilities. Detailed analysis of referral patterns and the specialized services offered by these hospitals will need to be conducted to understand the variations. This will help to explain the reasons behind the mortality contrasts.

Page 9:

86: regarding level of education, only three (3) patients were uneducated (no formal education), all of whom died within the 12 months follow up, obviously making it 100%; this is too small a number to even mention!!! 

Response: Well noted, we corrected ‘Education by excluding this category’, which led to slight changes in the Hazard ratios. 

Page 10:

Comorbidities: saying that 21 patients had no comorbidities; this is from a clinician's point of view very unlikely and hard to believe. 

Was HIV status known in the cohort or undetermined or ignored? This would have been very valuable info. Comorbidity was not significant: Variable for ‘comorbidities’ was available, and 96% (131/137) of the participants had other underlying conditions. However, this variable was not eligible for further analysis since its p value was >0.25 at bivariate analysis.

Response: The data on cormobidities was taken from the patients notes. this may have affected this information among the patients

HIV data: HIV data were excluded from the manuscript due to a high proportion of missing information (71 cases – 52%). Among these cases, 69.01% were deceased, indicating potential bias in the analysis if imputed or incomplete data were used.

The outcomes/survival times according to payer categories don't make sense as the non-paying ones are better off (but these are older pensioners etc.). The privately paying patients had poorer survival, but this may be related to the fact that these were sicker patients with no choice but biting the "private" bill? People in this category will not go to hospital for minor strokes. Patients staying with parents/relatives have better survival because they are probably younger... The inclusion of hospital bill payment modalities is crucial as it sheds light on economic barriers a

---

## [Decision Letter · Decision Letter 1]

2 Oct 2024

PONE-D-24-12258R1Factors influencing survival outcomes in patients with stroke at three tertiary hospitals in Zimbabwe: A 12-month longitudinal studyPLOS ONE

Dear Dr. Kaseke,

Thank you for submitting your manuscript to PLOS ONE. After careful consideration, we feel that it has merit but does not fully meet PLOS ONE’s publication criteria as it currently stands. Therefore, we invite you to submit a revised version of the manuscript that addresses the points raised during the review process. Please submit your revised manuscript by Nov 16 2024 11:59PM. If you will need more time than this to complete your revisions, please reply to this message or contact the journal office at plosone@plos.org. Please include the following items when submitting your revised manuscript:A rebuttal letter that responds to each point raised by the academic editor and reviewer(s). You should upload this letter as a separate file labeled 'Response to Reviewers'.A marked-up copy of your manuscript that highlights changes made to the original version. You should upload this as a separate file labeled 'Revised Manuscript with Track Changes'.An unmarked version of your revised paper without tracked changes. You should upload this as a separate file labeled 'Manuscript'.

We look forward to receiving your revised manuscript.

Kind regards,

Ryan G Wagner, MSc(Med), MBBCh, PhD

Academic Editor

PLOS ONE

**Additional Editor Comments:**

I would ask the authors to carefully review and directly respond to each point raised by the reviewers. 

Reviewers' comments:

Reviewer's Responses to Questions

**Comments to the Author**

1. If the authors have adequately addressed your comments raised in a previous round of review and you feel that this manuscript is now acceptable for publication, you may indicate that here to bypass the “Comments to the Author” section, enter your conflict of interest statement in the “Confidential to Editor” section, and submit your "Accept" recommendation.

Reviewer #1: All comments have been addressed

Reviewer #2: (No Response)

Reviewer #3: (No Response)

2. Is the manuscript technically sound, and do the data support the conclusions?

Reviewer #1: Yes

Reviewer #2: Partly

Reviewer #3: Partly

3. Has the statistical analysis been performed appropriately and rigorously? 

Reviewer #1: Yes

Reviewer #2: I Don't Know

Reviewer #3: Yes

4. Have the authors made all data underlying the findings in their manuscript fully available?

Reviewer #1: Yes

Reviewer #2: Yes

Reviewer #3: Yes

5. Is the manuscript presented in an intelligible fashion and written in standard English?

Reviewer #1: Yes

Reviewer #2: Yes

Reviewer #3: Yes

6. Review Comments to the Author

Reviewer #1: (No Response)

Reviewer #2: The authors have addressed a good many of the points raised in the first set of comments.

I particularly acknowledge the comment on the exclusion of HIV status data from this paper; please may the authors include this in the text as a point of notice, since HIV status and stroke do have a correlation and Zimbabwe is a high prevalence environment thus HIV is to be considered as a risk factor.

My main issue has still not been addressed, yet I do appreciate some of the comments in the authors' response; I copy here directly:

The outcomes/survival times according to payer categories don't make sense as the non-paying ones are better off (but these are older pensioners etc.). The privately paying patients had poorer survival, but this may be related to the fact that these were sicker patients with no choice but biting the "private" bill? People in this category will not go to hospital for minor strokes. Patients staying with parents/relatives have better survival because they are probably younger... The inclusion of hospital bill payment modalities is crucial as it sheds light on economic barriers and financial stressors that influence recovery, as discussed in the study introduction & discussion). This is often overlooked, particularly within Zimbabwean contexts, and may similarly pertain to other settings worldwide.

Here is the crux of the paper and my main criticism which can be allayed by proper/different statistics being applied:

The authors seem to find associations by the Gompertz AFT modelling statistics because Cox bivariate regression had failed to show significant correlations between the variables and time outcomes. The authors derive correlations based on the AFT stats, but seem to mix up association and causation: the fact that people living with parents/relatives have better outcomes than people living in their own homes may be related to the fact that the former are younger than the latter... These patients have a better outcomes because their strokes might be milder and they can "afford" to seek hospitalisation for minor strokes as opposed to others.

While I do accept the employment of the Gompertz AFT modelling statistics for the reasons explained by the authors, may some discussion please be devoted to the above.

Secondary education is better than primary education, but also better than tertiary education for stroke outcomes? Please may you read through the above comments again and reply accordingly in the manuscript with the arguments raised.

Reviewer #3: Authors have addressed all points raised on my initial review except one key issue. Authors should please provide a Table that compares participants who were lost to the follow-up versus those who completed the study versus those who died. For those who were lost, are their demographic, clinical and stroke severity indicators significantly similar or dissimilar to those who completed the study vs those who died? This is an important analysis given the significant attrition reported in your study. Thanks

7. PLOS authors have the option to publish the peer review history of their article (what does this mean?). If published, this will include your full peer review and any attached files.

Reviewer #1: No

Reviewer #2: **Yes: **Andre Mochan

Reviewer #3: **Yes: **Fred Stephen SARFO

---

## [Author Response · Author response to Decision Letter 1]

16 Oct 2024

REVIEWER COMMENTS 

Reviewer #2 comment:

The authors have addressed a good many of the points raised in the first set of comments.

I particularly acknowledge the comment on the exclusion of HIV status data from this paper; please may the authors include this in the text as a point of notice, since HIV status and stroke do have a correlation and Zimbabwe is a high prevalence environment thus HIV is to be considered as a risk factor.

RESPONSE: HIV data: Analysis by HIV status was not performed due to a significant proportion of participants with missing HIV status data (71 cases - 52%), but remained included in the analysis of other variables. The exclusion of HIV status was necessary to avoid misleading conclusions, as 69.01% of the cases with missing HIV data were deceased, which could have introduced bias if incomplete or imputed data had been used. – Added under study variables: line 137-141

The exclusion of analysis by HIV status, due to the large proportion of participants with missing HIV status data, is a limitation of this study. Future studies should focus on collecting more complete HIV data and analyse its potential impact on stroke outcomes, as HIV infection may significantly influence patient prognosis and recovery. – Added under Strengths and limitations of the study and or findings: line 450-453

Reviewer #2 comments: My main issue has still not been addressed, yet I do appreciate some of the comments in the authors' response; I copy here directly: 

RESPONSE: We thank the reviewer for the thoughtful feedback and for highlighting critical aspects of the study that require further clarification and discussion. We have carefully considered the comments, and we would like to provide a detailed response to address the points raised below.

Reviewer #2 comment: The outcomes/survival times according to payer categories don't make sense as the non-paying ones are better off (but these are older pensioners etc.). The privately paying patients had poorer survival, but this may be related to the fact that these were sicker patients with no choice but biting the "private" bill? People in this category will not go to hospital for minor strokes. 

RESPONSE: Survival Times according to payer categories: We acknowledge the reviewer’s observation that the survival times for non-paying patients appeared to be better, while privately paying patients exhibited poorer survival outcomes. This apparent discrepancy may indeed reflect differences in the severity of illness rather than solely the payment modality. Privately paying patients might be more likely to seek care only when their condition is severe, leading to worse outcomes. On the other hand, non-paying patients, including older pensioners, may have better access to public healthcare services and seek medical attention for milder conditions, which could result in better survival outcomes.

Discussion section revised to add the following, line 414-430: The observed differences in survival times across Hospital bill payment modalities may reflect not only economic disparities but also differences in health-seeking behaviors and illness severity at the time of hospital admission. Non-paying patients, which include social welfare, pensioners, retirees, and those older than 65, likely have earlier access to public healthcare services, potentially seeking care for less severe conditions, which could explain their better survival outcomes. Conversely, cash paying patients may delay seeking care due to financial barriers, only presenting at hospitals when their condition becomes critical. This difference in access to timely care likely contributes to the poorer outcomes seen in the privately paying group. This demonstrates the impact of economic barriers on health outcomes, particularly in contexts like Zimbabwe, where financial constraints may limit access to healthcare. The relationship between healthcare payment modalities and survival outcomes warrants further investigation, as it reflects broader socioeconomic inequalities that can significantly influence recovery and long-term survival.

Reviewer #2 Comment: The authors derive correlations based on the AFT stats, but seem to mix up association and causation: the fact that people living with parents/relatives have better outcomes than people living in their own homes may be related to the fact that the former are younger than the latter... These patients have a better outcomes because their strokes might be milder and they can "afford" to seek hospitalization for minor strokes as opposed to others. 

RESPONSE: Living with parents/relatives and better outcomes: We agree with the reviewer that patients living with parents or relatives tend to have better survival outcomes, potentially due to their younger age and less severe strokes. Younger patients may be more resilient and have better recovery prospects, and they may also be more likely to seek care for milder strokes. While our model identified this association, we acknowledge that the relationship between living arrangements and stroke outcomes may be influenced by factors such as age and stroke severity, which warrant careful consideration in the interpretation of our findings.

Discussion section revised to add the following, line 385-397: The improved outcomes associated with living with family may also be influenced by factors such as younger age and fewer comorbidities among these patients, making them more likely to recover from stroke. In addition, the support provided by family members likely facilitates quicker access to care, better adherence to treatment, and more effective management of overall health. In contrast, patients living independently may delay seeking treatment, contributing to poorer outcomes. While our analysis identified this association, it is important to acknowledge the role that age and stroke severity play in shaping these results. Future research should explore on the impact of family support on stroke recovery, especially in resource-limited settings where informal caregiving is a critical component of patient care.

Reviewer # 2 Comment: Secondary education is better than primary education, but also better than tertiary education for stroke outcomes? Please may you read through the above comments again and reply accordingly in the manuscript with the arguments raised. 

RESPONSE: Education Levels and stroke outcomes: Regarding the relationship between education levels and stroke outcomes, we recognize the unexpected result where secondary education appeared to be associated with better outcomes than both primary and tertiary education. We propose that this could be explained by several factors, including socioeconomic status, health-seeking behavior, and lifestyle differences between individuals with secondary and tertiary education. Those with secondary education may occupy a socioeconomic position that affords them better access to healthcare and healthier lifestyles, while individuals with tertiary education may face occupational stress and lifestyle factors that negatively affect their health.

Discussion section revised to add the following, line 365-375: Interestingly, the better outcomes associated with secondary education, compared to both primary and tertiary education, may be linked to socioeconomic and lifestyle factors. Individuals with secondary education likely occupy a middle socioeconomic tier, which provides better access to healthcare and healthier lifestyles. In contrast, those with tertiary education may face higher levels of occupational stress, contributing to risk factors like hypertension or sedentary behaviour, while individuals with primary education may lack health literacy or access to adequate healthcare, leading to poorer outcomes. This pattern reflects the complex ways in which education interacts with socioeconomic status and health behaviors to influence health outcomes. Understanding these dynamics is critical, particularly in low-resource settings where disparities in education and healthcare access can significantly affect disease outcomes.

Reviewer 2 comment: While I do accept the employment of the Gompertz AFT modelling statistics for the reasons explained by the authors, may some discussion please be devoted to the above. 

RESPONSE: Clarification on the use of the Gompertz AFT model and misinterpretation of causation: We appreciate the reviewer’s acceptance of the Gompertz AFT model. As mentioned in our previous responses, the selection of the Gompertz AFT model was based on thorough diagnostics, which indicated that it provided the best fit for the data. While significant associations were observed, we agree that it is crucial to distinguish between association and causation. The associations identified by the AFT model should not be interpreted as direct causal relationships but rather as indications of patterns within the data. We have been mindful to frame our findings accordingly and will ensure this is clearly articulated in the manuscript.

Discussion section revised to add the following, line 437-444: While the Gompertz AFT model identified significant associations between the observed socioeconomic and demographic factors and stroke outcomes, it is essential to emphasize that these associations do not imply direct causation. For example, the observed better outcomes for patients living with parents or relatives may be linked to their younger age or less severe strokes, rather than the living arrangement itself being a direct cause of improved survival. The model was selected based on its superior fit for the data compared to other models; however, like any statistical model, it identifies relationships that need to be considered in the broader clinical and social context. The intent behind employing the Gompertz AFT model was to provide a more precise analysis of time-to-event data, rather than to establish definitive cause-and-effect links.

Reviewer #3: Major comments 

1. Authors have addressed all points raised on my initial review except one key issue. 

Authors should please provide a Table that compares participants who were lost to the follow-up versus those who completed the study versus those who died. 

For those who were lost, are their demographic, clinical and stroke severity indicators significantly similar or dissimilar to those who completed the study vs those who died? This is an important analysis given the significant attrition reported in your study. Thanks 

RESPONSE: Table 2 with interpretations added: line 235-254

---

## [Decision Letter · Decision Letter 2]

22 Nov 2024

Factors influencing survival outcomes in patients with stroke at three tertiary hospitals in Zimbabwe: A 12-month longitudinal study

PONE-D-24-12258R2

Dear Dr. Kaseke,

We’re pleased to inform you that your manuscript has been judged scientifically suitable for publication and will be formally accepted for publication once it meets all outstanding technical requirements.

Kind regards,

Ryan G Wagner, MSc(Med), MBBCh, PhD

Academic Editor

PLOS ONE

Additional Editor Comments (optional):

Reviewers' comments:

Reviewer's Responses to Questions

**Comments to the Author**

1. If the authors have adequately addressed your comments raised in a previous round of review and you feel that this manuscript is now acceptable for publication, you may indicate that here to bypass the “Comments to the Author” section, enter your conflict of interest statement in the “Confidential to Editor” section, and submit your "Accept" recommendation.

Reviewer #2: All comments have been addressed

2. Is the manuscript technically sound, and do the data support the conclusions?

Reviewer #2: Yes

3. Has the statistical analysis been performed appropriately and rigorously? 

Reviewer #2: Yes

4. Have the authors made all data underlying the findings in their manuscript fully available?

Reviewer #2: Yes

5. Is the manuscript presented in an intelligible fashion and written in standard English?

Reviewer #2: Yes

6. Review Comments to the Author

Reviewer #2: (No Response)

7. PLOS authors have the option to publish the peer review history of their article (what does this mean?). If published, this will include your full peer review and any attached files.

Reviewer #2: **Yes: **Andre Mochan

---

## [Editor Report · Acceptance letter]

9 Dec 2024

PONE-D-24-12258R2 

PLOS ONE

Dear Dr. Kaseke, 

I'm pleased to inform you that your manuscript has been deemed suitable for publication in PLOS ONE. Congratulations! Your manuscript is now being handed over to our production team.

Kind regards, 

on behalf of

Prof. Ryan G Wagner 

Academic Editor

PLOS ONE